# Sodium Hyaluronate Nanocomposite Respirable Microparticles to Tackle Antibiotic Resistance with Potential Application in Treatment of Mycobacterial Pulmonary Infections

**DOI:** 10.3390/pharmaceutics11050203

**Published:** 2019-05-01

**Authors:** Irene Rossi, Francesca Buttini, Fabio Sonvico, Filippo Affaticati, Francesco Martinelli, Giannamaria Annunziato, Diana Machado, Miguel Viveiros, Marco Pieroni, Ruggero Bettini

**Affiliations:** 1Food and Drug Department, University of Parma, Parco Area delle Scienze 27/A, 43124 Parma, Italy; irene.rossi@studenti.unipr.it (I.R.); francesca.buttini@unipr.it (F.B.); fabio.sonvico@unipr.it (F.S.); filippo.affaticati@studenti.unipr.it (F.A.); Francesco.Martinelli@aptuit.com (F.M.); giannamaria.annunziato@unipr.it (G.A.); marco.pieroni@unipr.it (M.P.); 2Interdipartmental Center for Innovation in Health Products, BIOPHARMANET TEC, University of Parma, Parco Area delle Scienze 27/A, 43124 Parma, Italy; 3Global Health and Tropical Medicine, GHTM, Instituto de Higiene e Medicina Tropical, IHMT, Universidade Nova de Lisboa, UNL, 1349-008 Lisbon, Portugal; diana@ihmt.unl.pt (D.M.); MViveiros@ihmt.unl.pt (M.V.)

**Keywords:** nanoparticles, dry powder inhaler, sodium hyaluronate, mycobacteria, antibiotics, efflux pump inhibitor

## Abstract

Tuberculosis resistant cases have been estimated to grow every year. Besides *Mycobacterium tuberculosis*, other mycobacterial species are responsible for an increasing number of difficult-to-treat infections. To increase efficacy of pulmonary treatment of mycobacterial infections an inhalable antibiotic powder targeting infected alveolar macrophages (AMs) and including an efflux pump inhibitor was developed. Low molecular weight sodium hyaluronate sub-micron particles were efficiently loaded with rifampicin, isoniazid and verapamil, and transformed in highly respirable microparticles (mean volume diameter: 1 μm) by spray drying. These particles were able to regenerate their original size upon contact with aqueous environment with mechanical stirring or sonication. The in vitro drugs release profile from the powder was characterized by a slow release rate, favorable to maintain a high drug level inside AMs. In vitro antimicrobial activity and ex vivo macrophage infection assays employing susceptible and drug resistant strains were carried out. No significant differences were observed when the powder, which did not compromise the AMs viability after a five-day exposure, was compared to the same formulation without verapamil. However, both preparations achieved more than 80% reduction in bacterial viability irrespective of the drug resistance profile. This approach can be considered appropriate to treat mycobacterial respiratory infections, regardless the level of drug resistance.

## 1. Introduction

Tuberculosis (TB), a bacterial infection caused by *Mycobacterium tuberculosis* (Mtb), is a major communicable disease. According to the World Health Organization (WHO), 10 million new TB cases were estimated in 2017, and 1.3 million people died among HIV-negative people. 9% of all the new cases occurred in HIV infected patients, among which TB is the leading cause of death, especially in Africa. In 2017, 558,000 new cases of multidrug (MDR) and rifampicin resistance (RR) TB have been estimated, further worsening an already drastic scenario [1]. Apart from Mtb, other mycobacterial species are getting a great deal of attention for public health in many geographical regions, as they cause very difficult-to-treat infections in immunocompromised as well as in immunocompetent patients. Pulmonary infection, lymphadenitis, skin and soft tissue infections are the most common expressions of the disease. Concerning pulmonary infections, nontuberculous mycobacteria (NTM) are, to some extent, similar to Mtb, as they also start by infecting macrophages [2]. However, whereas Mtb interactions with the immune system have been extensively studied, in particular, with respect to the role played by alveolar macrophages (AMs), limited information about the immune response against NTM is available [3].

Mtb enter the human lung through the inhalation of suitably small bacilli-carrying droplets (1–5 µm) that can deposit in the alveolar space [4]. Alveolar macrophages provide the first line of defense to pulmonary infection in virtue of their intracellular bacterial killing activity and subsequent antigen presentation to lymphocytes. However, in the case of mycobacterial infections, AMs fail to fulfil their immunological function and represent the environment where mycobacteria survive, reside and proliferate [5]. This intracellular survival is well documented for Mtb [6,7,8,9] and is now better described and understood for NTM [8,9]. 

Antituberculosis drugs are divided in first-line agents (i.e., isoniazid, rifampicin, ethambutol, pyrazinamide), delivered orally for long periods, up to 6 to 8 months, and second-line drugs (aminoglycosides and fluoroquinolones), mostly administered intravenously or intramuscularly [10].

In addition to first-line antituberculosis drugs, the adjuvant treatment with efflux transporters inhibitors [11], such as verapamil or thioridazine [11,12,13] has demonstrated to be beneficial in increasing the intracellular concentrations of antibacterials [11,12,13,14,15,16] shortening the duration of treatment [12,17], and reducing the onset of drug resistance. Resistance to rifampicin and isoniazid acquired by mutation establishes MDR strains. However, intrinsic mechanism, i.e., expression of efflux pumps, reduction of cell permeability and drug degradation, are the first steps of diminished sensitiveness against a specific antibiotic [13,14,17]. It has been recently observed that verapamil is able to block activity of the of macrophage-induced bacterial efflux pumps, resulting in an enhanced drug concentration within the AMs and in a rapid killing of entrapped mycobacteria without harming the host cell [13]. In addition, a recent paper reported that verapamil may disrupt bacterial membrane function besides acting on AMs efflux pumps [18]. However, oral therapeutic doses of the efflux pump inhibitors cause significant side effects that impair their clinical use [19]. 

The difficult accessibility to the pharmacological target and the peculiar capability of Mtb to slow-down its metabolism and survive in a dormant state makes treatment of tuberculosis a challenging task [20]. Because the lungs are the primary site of infection for Mtb, the interest in a local administration to the relevant organ (e.g., effective pulmonary delivery of antibiotics) has grown during the last decades [21]. Moreover, several studies have reported various antibiotic delivery strategies targeting AMs [22] to increase efficacy, reduce resistance and, at the same time, lower systemic drug exposure, and consequently side effects.

The aerodynamic particle size of aerosols intended to be administered to the lung has to be lower than 5 μm [23]. On the other hand, to increase the concentration of antibiotics penetrating the infected AMs, it has been reported that particles should be the range of 0.20–0.55 µm since they are more likely to be subjected to phagocytosis by AMs [21,22,23,24,25,26,27,28]. However, aerosol particles with such aerodynamic diameter (<0.5 μm) are more prone to be exhaled and submicron-sized powders are known to be particularly cohesive and display extremely poor flow and aerosolization properties. Thus, a powder dosage form with such a particle size distribution would be difficult to handle and, in any case, would not be deposited efficiently into the lungs. In reality, two feature are required in order to optimize the antituberculosis therapy: high drug loading of the powder to be inhaled and the enhancement of alveolar macrophage uptake [21,29]. This latter feature may be attainable through the use of specific chemical and physicochemical signals related to the nature of the material used for particle construction. While there is abundant literature describing the effect of particle size on AM capture, the influence of the chemical nature of the vehicle has only recently attracted specific interest [30,31].

In this respect it has been reported that low molecular weight hyaluronic acid sodium salt (HA) is capable to be efficiently internalized by AMs via CD44 surface receptors [32]. Furthermore, low molecular weight HA is able to polarize macrophages at M1 phenotype [33], promoting an inflammatory response. This macrophage activation could be exploited to further improve the treatment of infectious diseases such as TB [34,35].

Based upon the convergence of all these findings, the present work proposes a novel formulation approach for the treatment of Mtb infections potentially suitable to overcome antibiotic resistance. The specific aim of this research was to design and develop highly respirable particles with enhanced AMs uptake and able to provide increased intracellular antituberculosis drug concentrations through a local impairment of drug efflux. This goal was pursued through the generation of micron-sized aggregates of nanoparticles with size (300–500 nm) and composition (low molecular weight HA) suitable to enhance their capture by alveolar macrophages. These particles were loaded with two different antibiotics, rifampicin and isoniazid, and one efflux pump inhibitor, verapamil. 

A specific feature of the proposed approach is the fact that the formulation targets AMs along with the mycobacteria *per se* and includes an efflux pump inhibitor to further increase the concentration of the antibiotics inside the mycobacteria and simultaneously promote intra-macrophagic killing of the mycobacteria. Therefore, this approach is expected to work regardless of the antibiotic sensitiveness profile and the mycobacterial species.

## 2. Materials and Methods

### 2.1. Materials

Antibiotics, isoniazid and rifampicin, were purchased by Alfa Aesar GmbH & Co. KG (Karlsruhe, Germany) and Sanofi-Aventis S.p.A. (Milan, Italy), respectively. Verapamil hydrochloride was bought by Hangzhou Dayangchem Co. Ltd. (Hangzhou City, China). Low molecular weight HA (PrymalHyal 50, average *M*_W_ 29.5 kDa) was obtained from Soliance (Pomacle, France). For dissolution experiments phosphate buffer saline (PBS) at pH 7.4 was prepared according to Ph. Eur. 9.0 Ed., using monobasic potassium phosphate and sodium hydroxide (A.C.E.F., Fiorenzuola D’Arda, Italy), while dialysis membranes (cut-off *M*_W_ 14 kDa) were obtained from Sigma-Aldrich (Saint Louis, MO, USA). All solvents used were of analytical grade and ultrapure water (0.055 μS/cm, TOC 1 ppb) was obtained by reverse osmosis (Purelab Pulse + Flex ultrapure water, Elga-Veolia, Milan, Italy). 

### 2.2. Methods

#### 2.2.1. Turbidimetry

A turbidimetric analysis was employed to set-up the antisolvent process in water/ethanol mixtures finalized at obtaining a HA colloidal nanosuspension. Low molecular weight HA was tested at 7 different starting concentrations: 1.0, 1.5, 2.0, 2.5, 3.0, 3.5 and 4.0% *w*/*v*, in a 25 mL of a hydroalcoholic mixture 50:50. The starting solutions was prepared by dissolving HA in 12.5 mL of water and subsequently adding 12.5 mL of ethanol (95% *v*/*v*). Thereafter, ethanol (95% *v*/*v*) was added dropwise to the solution as antisolvent and the system was left to equilibrate for 5 min under stirring at 200 rpm. Transmittance value was recorded (Jasco V-530, Tokyo, Japan) at 400 nm after each addition of antisolvent until a sharp drop of transmittance (below 10%) was obtained. Water:ethanol at the same ratio was used as reference blank solution. The transmittance was measured in flow-through cuvettes (1 cm optical pathlength) connected via a peristaltic pump (Esapump, Advanced Products, Milan, Italy) to the test solution.

#### 2.2.2. Triple Drug Combination HA Nanoparticles-Containing Respirable Powder Produced by Spray Drying

A powder containing HA at low molecular weight and the three drugs, i.e., rifampicin, isoniazid and verapamil, was produced employing a mini spray dryer B-290 (BÜCHI, Flawil, Switzerland) starting from a nanosuspension obtained with the above-mentioned antisolvent process from a hydroalcoholic solution containing the three drugs along with HA (0.59% *w*/*v*). The drying parameters were set as follow: inlet temperature (90 °C), drying air flow rate (750 L/h), aspiration (35 m^3^/h), solution feed rate (3.0 mL/min) and spray nozzle diameter 0.7 mm [36]. The outlet temperature recorded was 56 °C. To produce the starting nanosuspension 210 mg of HA and 110 mg of isoniazid were dissolved in 30 mL of purified water, while 210 mg of rifampicin and 70 mg of verapamil HCl were dissolved in 70 mL ethanol. The latter solution was added to the aqueous one under continuous magnetic stirring at 200 rpm, and contemporary homogenization with Ultra-TURRAX^®^ TP 18/10 at 10,000 rpm for 15 min (IKA Werke GmbH & Co., Staufen im Breisgau, Germany) in order to allow the nanoparticle formation.

The same production process was adopted to prepare a similar powder but without verapamil that was used for comparative purpose in the in vitro and ex vivo experiments (see Section 2.2.11 and Section 2.2.12). 

#### 2.2.3. Spray Dried Powder Drug Loading Efficiency and Drug Content

The amount of drugs loaded in the nanoparticles was evaluated in the starting nanosuspension as well as in the dried powder. About 30 mg of powder were accurately weighed, dispersed in 5 mL of ultrapure water and placed in an ultrasonic bath (8510 E-DTH, Branson Ultrasonics Co., Danbury, CT, USA) for 7 min in order to obtain a homogenous suspension. Samples of 500 μL of both starting nanosuspension and suspension obtained from the powder, were filtered using Vivaspin^®^ 500 centrifugal concentrators (Sartorius, Göttingen, Germany) following the instructions of the manufacturer using a centrifuge (Model D3024, Scilogex, Rocky Hill, CT, USA) operated at 20,000× *g* for 15 min at room temperature. The obtained filtrates, which contained the drugs in solution and hence not entrapped in the nanoparticles, were analyzed by HPLC. 

The total drugs content in the spray dried powder was evaluated by dissolving 5 mg of powder in 100 mL of a solution water: acetonitrile 50:50 *v*/*v*. The obtained solutions were analyzed by HPLC. 

#### 2.2.4. HPLC Analysis

A specific High Performance Liquid Chromatography with UV-detection (HPLC-UV) method for the quantification of isoniazid, rifampicin and verapamil HCl was developed modifying the method reported by Chan et al. [37]. An Agilent 1200 Series HPLC-UV (Agilent Technologies, Santa Clara, CT, USA) equipped a C18 μBondapak, 10 μm, 3.9 × 300 mm column (Waters Chromatography Ltd., Dublin, Ireland) was used. 

Standard and samples were prepared in water:acetonitrile 50:50 *v*/*v*. Mobile phase was pumped in gradient elution using: (A) 0.05 M potassium phosphate buffer (KH_2_PO_4_) adjusted to pH 3 with diluted phosphoric acid (10% *v*/*v*); (B) methanol; (C) ultrapure water and (D) acetonitrile. The gradient adopted is reported in Appendix A. The injection volume was set at 10 μL, while flow rate and wavelength were modified during the analysis (see Appendix A), in order to detect the three drugs. Retention time was 3.5 min for isoniazid, 20 min for rifampicin and 40.5 min for verapamil HCl. Linearity of the responses was assessed between 0.001 and 0.1 mg/mL (*R*^2^ = 0.999) for each drug. The limit of detection (LOD) and the limit of quantification (LOQ) were between 1–2 and 3–4 µg/mL, respectively, for all drugs.

#### 2.2.5. Particle Size Distribution

Dynamic Light Scattering (DLS, ZetaPALS^®^, Brookhaven, NY, USA) was used to measure the average particle size of the nanosuspension at 25 °C before spray drying and to investigate the de-aggregation kinetics of the spray dried powder when dispersed in ultrapure water under two different conditions: upon ultrasounds application for 1, 5, 10 and 15 min (Ultrasonic Bath, USC-T, VWR, Radnor, PA, USA), or after magnetic stirring (200 rpm) at room temperature for 30, 50, 60 and 75 min. 

The spray dried powder was characterized in terms of particle size distribution using laser diffraction (Spraytec^®^, Malvern, UK). About 5 mg of powder were suspended in 5 mL of isopropanol and sonicated for 5 min. Instrument obscuration threshold was set at 5%. Data were expressed as volume diameter of 10th (*D*_v,10_), 50th (*D*_v,50_) and 90th (*D*_v,90_) percentile of the particle population and as Span value [(*D*_v,90_ − *D*_v,10_)/*D*_v,50_]. 

All particle size measurements were carried out in triplicate.

#### 2.2.6. Thermal Analysis and Solid-State Characterization

Thermogravimetric analysis (TGA/DSC1 STARe System, Mettler Toledo, Milan, Italy) was carried out by placing about 5 mg of powder samples in a 70 μL alumina pans with a pierced cover. Samples were heated under a flux of dry nitrogen (100 mL min^−1^) at 10 °C min^−1^ in the 25–150 °C temperature range. Each analysis was performed in triplicate. 

Differential Scanning Calorimetry (DSC) was adopted to evaluate the solid-state of raw materials and of the spray dried powder. Measurements were performed on an Indium calibrated (onset of melting *T*_m_ = 157.1 °C, enthalpy of melting ΔH_m_ = 27.84 J g^−1^) Mettler DSC 821e equipment (Mettler Toledo, Milan, Italy) operated using STARe software (Version 11, Mettler Toledo, Milan, Italy, 2014). DSC traces were recorded by placing accurately weighed quantities (5–11 mg) in a 40 µL Aluminum pan which was then sealed and double pierced. Scans were performed between 25 and 250 °C with a heating rate of 5 °C min^−1^ under a flux of dried nitrogen (100 mL min^−1^). Each powder sample was analyzed at least in duplicate. Data relevant to the observed thermal events were reported as onset and peak temperatures.

Crystalline state characterization was also carried out by Powder X-ray Diffraction (PXRD, Miniflex, Rigaku, Japan). The powder sample (about 300 mg) was placed in a sample holder and flattened using a glass slide. Radiation CuKα of 30 kV, scan speed of 0.05 °C min^−1^ and a range of scan between 2° and 35° were used. 

#### 2.2.7. Microscopy

Microparticles and nanoparticles morphology were studied by Scanning Electron Microscopy (SEM, SUPRA 40, Carl Zeiss NTS GmbH, Jena, Germany) and Transmission Electron Microscopy, (TEM, JEM 2200-FS microscope, JEOL Ltd., Tokyo, Japan) respectively. 

For SEM analysis, powders were deposited on adhesive black carbon tabs pre-mounted on Aluminum stubs and exposed to a gold metallization process to deposit a gold film of 60 nm. The microscope was operated after 30 min of depressurization under high vacuum conditions (1.33 × 10^−2^ Pa) with an accelerating voltage of 1.5 kV. 

The morphological analysis of the nanoparticles was carried out using TEM operated at 80 kV. Nanoparticles were recovered from the dried powder and dispersed in ultrapure water (1 mg/mL) and placed in an ultrasonic bath for 5 min. For sample preparation, a drop of the suspension was deposed on formvar/carbon coated copper grids (300 Mesh, Electron Microscopy Sciences, Hatfield, PA, USA). After 10 min the excess sample was gently dried with filter paper and the sample was stained using a drop of UranyLess (Electron Microscopy Sciences, Hatfield, PA, USA) for 90 s. Then, the grid was rapidly dipped in particle free ultrapure water to remove excess staining. 

#### 2.2.8. Aerodynamic Performance

The aerodynamic performance of powders was investigated using a Fast Screening Impactor (FSI, Copley scientific Ltd., Nottingham, UK). This is an abbreviated impactor that permits to divide the powder emitted from a device in two fractions: the first one composed of particles with an aerodynamic diameter >5 μm, which is collected inside the Coarse Fraction Collector (CFC), and the second one constituted of particles with an aerodynamic diameter <5 μm, collected in the Fine Fraction Collector (FFC). In order to carry out the aerodynamic performance experiment, 5 mg of powder (corresponding to 1.75 mg of HA, 1.75 mg of RIF, 0.9 mg of INH, 0.6 mg of VER) were loaded manually in a hypromellose Quali-V-I capsule size 3 (Qualicaps^®^, Madrid, Spain) and aerosolized using two different devices: RS01^®^ (RPC, Plastiape, Osnago, Italy) and Turbospin^®^ (PH&T, Milan, Italy). A single capsule was discharged inside the impactor for each test. The entire system was connected to a vacuum pump (Mod. 1000, Erweka GmbH, Heusenstamm, Germany) which created the air flow necessary to aerosolize the powder and entrain it in the FSI. The flow rate used during each test was adjusted, according to current USP monograph, with a critical flow controller (TPK, Copley Scientific, Nottingham, UK) in order to produce a pressure drop of 4 kPa over the inhaler. Thus, flow rates of 60 L/min for RS01^®^ and of 65 L/min for Turbospin^®^ were set before each experiment using a flow meter (DFM 2000 Copley Scientific, Nottingham UK). The device was activated and the vacuum applied for 4 s for RS01^®^ and for 3.7 s for Turbospin^®^, so that a total volume of 4 L of air was drawn through the inhaler during the experiment.

The aerodynamic performance was tested in triplicate and the samples were collected using water:acetonitrile mixture 50:50 *v*/*v*. The residual amount of the drugs still present in the capsule and in the device after aerosolization as well as the powder deposited in the rubber adaptor and in the induction port, were solubilized in 50 mL of the same solvent mixture. Ten mL of solvent mixture were put in the CFC insert before the aerosolization; successively another 20 mL were used to collect the powder inside CFC. Finally, the filter (glass fiber filter type A/E 76 mm diameter, Pall Corporation, New York, NY, USA) in the FFC was transferred in a glass crystallizer and washed with 20 mL of the solvent mixture; the crystallizer was then put in an ultrasonic bath for 5 min, in order to allow complete dissolution of the powder deposited on the filter. The solutions obtained were analyzed by HPLC.

The quantification of the drugs deposited in the impactor allowed for the calculation of the following aerodynamic parameters: the emitted dose (ED) was the amount of each drug which was emitted from the device namely, the amount of drug collected from the induction port (IP) to the filter (F). The emitted fraction (EF) was calculated as the ratio between the ED and the amount of powder loaded in the capsule. The fine particle dose (FPD) was the mass of each drug with aerodynamic diameter lower than 5 μm, i.e., the amount found in the FFC filter; the fine particle fraction (FPF) was calculated as the ratio between FPD and ED.

#### 2.2.9. Preliminary Stability Assessment

Stability of the spray dried powder was assessed by measuring the drugs content and aerodynamic performance [38] (RS01^®^) after storing about 100 mg of the powder in a 20 mL amber vial sealed with a clamped elastomeric stopper both for 1 month under ambient conditions and for 24 h at 50 °C. 

#### 2.2.10. Dissolution

The dissolution rate of the drugs from the spray-dried powder was investigated with a Franz type cell (surface 60 mm^2^) with a dialysis membrane (cut-off 14 kDa) placed between the donor and the receptor compartments. The test was carried out using 4.7 mL of phosphate buffer at pH 7.4 (Ph.Eur. 9.0 Ed.), for 40 h at 37 °C, under continuous magnetic stirring (350 rpm). In detail, the receptor was loaded with 4.2 mL of phosphate buffer and a dialysis membrane (cut-off MW 14 kDa) was put between the donor and receptor compartments. Then, 0.1 mL of phosphate buffer were added to wet the upper side of the membrane in the donor. The assembled system was equilibrated for 1 h at 37 °C. Successively, 5 mg of powder were carefully spread onto the dialysis membrane in the donor and 0.4 mL of phosphate buffer were added to wet the deposited powder. Samples (0.7 mL) were withdrawn from the receptor at fixed time. After each sampling (5, 10, 15 min and then about every hour for a total time of 40 h) the volume of the solution withdrawn was replaced with the same volume of fresh buffer, so as to keep the total volume of buffer inside the receptor constant. The amount of the three drugs dissolved were quantified by HPLC using the method described above.

#### 2.2.11. Minimum Inhibitory Concentrations of the Compounds against *M. tuberculosis* Strains

MIC determination of the compounds was conducted by the 96-well broth microdilution method using a tetrazolium microplate-based assay [39,40]. Strains tested included the *M. tuberculosis* H37Rv reference strain and two clinical drug resistant strains: Mtb1, a MDR strain, and Mtb2, an extensively drug resistant (XDR) strain. Strains were grown in Middlebrook 7H9 (MB7H9, Difco, Becton and Dickinson, Franklyn Lakes, NJ, USA) supplemented with 10% OADC (oleic acid, albumin, dextrose, catalase; Becton and Dickinson) at 37 °C until reaching an optical density (OD) of 0.8 at 600 nm. The inoculum was prepared by diluting the bacterial cultures in MB7H9/OADC to a final density of approximately 10^5^ cells/mL [41]. Aliquots of 0.1 mL of inoculum were transferred to each well of the plate that contained 0.1 mL of each compound at concentrations prepared from two-fold serial dilutions in MB7H9/OADC medium. Growth controls and a sterility control were included in each assay. The inoculated plates were sealed in plastic bags and incubated at 37 °C for 7 days. After this period, 10 µL of a MTT working solution was added to each well and the plates incubated overnight. The 10× MTT stock solution was prepared in ultrapure sterile water and used as working solution at 1:1 in 10% Tween 80 (*v*/*v*) (Sigma-Aldrich, Saint Louis, MO, USA). The bacterial viability was registered for each well based on the MTT color change and the MIC was defined as the lowest concentration of compound that totally inhibited bacterial growth (no color change) [39]. The assays were performed in triplicate. 

#### 2.2.12. Evaluation of the Intracellular Activity against *M. tuberculosis* Strains

##### Isolation of Human Monocyte-Derived Macrophages

Blood was collected from healthy donors and peripheral blood mononuclear cells were isolated by Ficoll-Paque Plus (GE Healthcare Biosciences, Piscataway, NJ, USA) density gradient centrifugation at 4 °C. The conduct of the experiments derived from peripheral blood from volunteer donors was consistent with the principles of the Declaration of Helsinki. Recruited volunteer blood donors were asked to give their informed consent to participate. The research did not involve any personal data collection and/or processing nor storage of the donated samples as per the institutional operational SOPs and guidelines for ex vivo studies, approved by the ethical committee of the IHMT/UNL (Lisbon, Portugal) under the scope of the project “Targeting efflux pumps in drug resistant *Mycobacterium tuberculosis*: an approach to prevent efflux-mediated resistance and boost therapy in active and latent tuberculosis”, Ref. PTDC/BIA-MIC/30692/2017, approved by Fundação para a Ciência e a Tecnologia (FCT, Portugal), 14 May 2018.

Monocytes were differentiated into macrophages during 7 days in macrophage medium containing RPMI-1640 medium with 10% fetal calf serum (FCS), 1% GlutaMAX, 1 mM sodium pyruvate, 10 mM HEPES at pH 7.4, 100 IU/mL penicillin and 100 µg/mL streptomycin (Gibco, Life Technologies, Gaithersburg, MD, USA), and 20 ng/mL M-CSF (Immunotools, Friesoythe, Germany) and incubated at 37 °C with 5% CO_2_. Fresh medium was added at day 4 post isolation [42]. 

##### Cytotoxicity Analysis

The effect of the encapsulated powder formulations and the free compounds on human monocyte-derived macrophages was evaluated using the AlamarBlue method (Molecular Probes, Thermo Fisher Scientific, Eugene, OR, USA) as previously described [43]. Briefly, 5 × 10^4^ cells were seeded in 96-well microplates and exposed to the compounds suspended in cell medium during 3 h, and 1, 3, 5 and 7 days at 37 °C in a 5% CO_2_ atmosphere. The SD powder A was prepared at 179 µg/mL and the SD powder B was prepared at 165 µg/mL. The suspensions were two-fold serial diluted to 11.2 and 10.3 µg/mL, respectively, and 100 µL of each dilution added to the corresponded well of the microplate containing HMDM. After each time point of exposure the media with the drugs were removed, replaced with 100 µL of fresh cell culture media containing AlamarBlue to a final concentration of 10% and the fluorescence read after overnight incubation at 37 °C and 5% CO_2_ to evaluate cell viability. Fluorescence was measured at an excitation of 540 nm and emission of 590 nm in a Synergy HT multi-mode microplate reader (Biotek Instruments, Winooski, VT, USA). 

##### Ex Vivo Assays

*M. tuberculosis* cultures were prepared in MB7H9/10% OADC supplement plus 0.05% tyloxapol (Sigma-Aldrich, Saint Louis, MO, USA). The human monocyte-derived macrophages were infected with *M. tuberculosis* strains (H37Rv reference strain and two clinical drug resistant strains: Mtb1, a MDR strain, and Mtb2, an XDR strain) at a multiplicity of infection (MOI) 1:1 and were allowed to uptake the bacteria for 3 h. Following the uptake, the cells were washed three times with PBS to remove extracellular bacteria. Afterwards, powder formulations suspended in cell medium and free compounds were added to the infected macrophages at the desired concentrations. At 3 h, day one and day three post-infection, cells were lysed with 0.05% Igepal (Sigma-Aldrich). Serial dilutions of the lysate were placed on Middlebrook 7H11 (Difco) supplemented with 10% OADC. Colony forming units were counted upon 21 days of incubation at 37 °C. The compositions of the powder formulations and free compounds tested were as follows: SD powder A (nanoparticles formulation, NPs)—concentration 89.5 µg/mL, with verapamil-rifampicin 31.3 µg/mL; isoniazid 16.1 µg/mL; verapamil 10.7 µg/mL (as baseline); low molecular weight HA 31.4 µg/mL;SD powder B (nanoparticles formulation NPs)—concentration 82.5 µg/mL, without verapamil-rifampicin 31.4 µg/mL (as baseline); isoniazid 19.8 µg/mL; low molecular weight HA 31.4 µg/mL;solution A (free solution mixtures, Non-NPs)—rifampicin, 30 µg/mL; isoniazid, 15 µg/mL; verapamil, 10 µg/mL (as baseline);solution B (free solution mixtures, Non-NPs)—rifampicin, 30 µg/mL (as baseline); isoniazid 17.6 µg/mL.

All work involving *M. tuberculosis* cultures were carried out at the BSL-3 facility of the Mycobacteriology laboratory of the IHMT/UNL of Lisbon (Lisbon, Portugal).

Statistical analysis was carried out using the Student’s *t*-test. A * *p* value < 0.05 was considered statistically significant and highly significant when ** *p* < 0.01 and *** *p* < 0.001 (two-tailed tested).

## 3. Results

### 3.1. Particle Production

Low molecular weight HA was dissolved in purified water and then, ethanol, the antisolvent agent, was added dropwise to produce a colloidal suspension. The effect of HA concentration and water: ethanol ratio on the HA nanoparticle formation were investigated by turbidimetry. HA water solutions (in the concentration range 1–4 % *w*/*v*) remained clear over a certain volume of added ethanol which depended upon the HA content in the solution. This volume was lower for the higher HA concentrations and was calculated from the inflection point of each transmittance vs. ethanol concentration curve (Figure 1). These inflection points were used to draw the phase equilibrium curve (Figure 2): on the left-hand side of the curve a limpid solution is obtained, whereas the right-hand side is relevant to particles progressively increasing their size. When hyaluronate concentration was higher, the amount of ethanol required to precipitate HA and obtain a turbid HA dispersion was lower. The mathematical function describing the diphasic equilibrium curve was a decreasing exponential curve representing the relationship between the final low molecular weight HA concentration in solution and its solubility limit in the water:ethanol mixture. In particular, a very low level of HA concentration in the mixture was clearly associated with a solution even at higher level of ethanol, while the suspension appeared by increasing the HA concentration at fixed ethanol percentage following a non-linear relation. A stable colloidal suspension was obtained in correspondence or in proximity of the curve representing the transition between the two phases [36].

This method had to be suitably modified to produce HA nanoparticles in presence of antituberculosis drugs, starting from a water solution of low molecular weight HA and isoniazid and an ethanol solution of rifampicin and verapamil HCl, because the equilibrium solubility of HA was altered in the presence of other solutes. In order to optimize the starting suspension in terms of total solid concentration and components ratio a preliminary formulation study was carried out, employing laser light scattering instead of turbidimetry (see Appendix A). The conditions chosen for nanoparticles preparation were the following: isoniazid 18%, rifampicin 35%, verapamil 12% and, finally, low molecular weight HA 35% *w*/*w*. The obtained nanoparticles presented an average hydrodynamic diameter of 480 and a polydispersity index of 0.2.

### 3.2. Powder Production, Thermal and Solid-State Analysis

The nanosuspension was then spray dried with a yield of 50.1%. Most of the powder remained adhered to the glass surfaces of cyclone and collector; this phenomenon could be attributed to the high residual water content of the powder (6.3% *w*/*w*, as evaluated by thermogravimetric analysis) and ascribed to the hydrophilic nature of HA that tends to entrap water in its structure [44]. The high content of water was confirmed by DSC analysis of the spray dried (SD) powder which presented a broad endothermic event between 25 and 130 °C (see Appendix A), due to evaporation of residual water bound to the polymer. This phenomenon was followed by an exothermic peak at high temperature (210–285 °C), ascribable to the thermal degradation of one or more components of the powder (likely HA).

Solid state analyses of the single components and of the SD powder was also carried out by PXRD (see Appendix A). 

The drug content after the production reflected the theoretical values (Table 1) with a standard deviation ≤0.03% indicating an even drug distribution in the powder. Similar results were observed after one month of storage at 25 °C, whereas, when the powder was stored at 50 °C for 24 h, isoniazid content significantly decreased, while only a slight degradation (well below 5%) was observed for rifampicin and verapamil.

### 3.3. Encapsulation Efficiency

The encapsulation efficiency data on the nanosuspension reported an entrapment efficiently higher than 50% for all the drugs. For rifampicin the encapsulation efficiency was 84.57 ± 3.47%. For verapamil the totality of the dose was encapsulated in the nanoparticles (100.00 ± 0.01%). Isoniazid was the one showed the lowest value (53.93 ± 0.82%). 

After the spray drying process the encapsulation percentages decreased for all the drugs: 48.75% ± 4.32 for isoniazid; 77.52% ± 0.11 for rifampicin and 80.45% ± 4.80 for verapamil. 

### 3.4. Aerodynamic Performance

The SD powder was further investigated in terms of particle size distribution by laser diffraction. It proved to be composed of microparticles with median volume diameter (*D*_v,50_) of 0.94 ± 0.15 μm, and a narrow monomodal distribution (Figure 3) as indicated by span value 1.79 ± 0.17. Although these diameters were geometric rather than aerodynamic, these data suggested that the powder may be particularly suitable for inhalation and deposition in the deep alveolar region (Figure 3). 

The aerodynamic performance was evaluated with two different devices (RS01^®^ and Turbospin^®^) using the Fast Screening Impactor (Figure 4). Turbospin^®^ was more efficient in emitting of the powder when compared to RS01^®^. Indeed, the total emitted dose of the three drugs with the Turbospin^®^ device was 2.24 ± 0.42 mg, corresponding 74.25 ± 8.20% of the loaded dose versus 1.62 ± 0.49 mg with RS01^®^, i.e., 50.40 ± 15.83% of the loaded dose. For both devices, the highest percentage of drugs was found in the Fine Fraction Collector, where the particles with an aerodynamic diameter <5 μm are collected. Furthermore, the deposition in the FFC was higher, although not significantly, for Turbospin^®^. Nevertheless, it is worth noting that the powder emitted from the RS01^®^ deposited almost entirely in the FFC (more than 90% of the ED for each drug vs. around 73% for the Turbospin^®^).

Table 2 summarizes the aerosolization performance of the three drugs with the two devices. In all cases the respirability was high, with FPF greater than 70% for all the drugs relative to the emitted dose). The Fine Particle Dose of the powder aerosolized with the two devices was not statistically different (*p* = 0.5–0.9), indicating that both devices presented the same de-aggregation efficiency despite the powder was constituted by very small particles, *D*_v,50_ < 1 μm, and contained a significant amount of residual moisture.

### 3.5. Preliminary Stability Assessment

In order to evaluate the powder stability, the measurement of the aerodynamic performance was repeated with RS01^®^ device, after one month of storage at room temperature and after a thermal stress test (24 h at 50 °C). The powder resulted to be very stable at room temperature with values of ED and FPD (Table 3) not significantly different (*p* > 0.2) from those obtained with the same inhaler at time 0. On the contrary, the storage at high temperature caused a significant (*p* < 0.02) decrease of the emitted and respirable dose compared to the values obtained initially (Table 3). Rifampicin was the drug reporting the major reduction in terms of respirable fraction (84% after 24 h at 50 °C and 91% after 1 month at 25 °C against a value of 94% recorded at time 0).

### 3.6. De-aggregation Kinetics

The de-aggregation kinetics of the microparticles were investigated by Dynamic Laser Scattering. Figure 5 reports the diameter of the particles in suspension obtained when the spray dried powder was re-suspended in ultrapure water. After 75 min of simple magnetic stirring, the average particle size of the dispersed particles reached a value comparable to that of the initial nanosuspension (300–400 nm). As expected, this time was considerably lower (15 min) when the re-suspended particles were submitted to ultrasounds.

### 3.7. Morphology Characterization

The morphology of micro- and nanoparticles was studied using SEM and TEM, respectively. SEM pictures of spray dried microparticles (Figure 6A) show irregularly shaped particles characterized by a slightly rough surface. The same powder was dispersed in water and placed in an ultrasonic bath for 15 min in order to allow de-aggregation of nanoparticles as described in the previous section. Figure 6B shows the typical structure of the observed nano-sized assemblies, clearly evidencing the presence of nanoparticles detaching from larger agglomerates.

### 3.8. Dissolution

Figure 7 shows the release rate of the three drugs from the SD powder. Isoniazid was completely released within 24 h: after 20 h, the amount dissolved was already attaining 80%. The release rate of rifampicin and verapamil was lower due to their relatively low solubility in water (2.5 mg/mL and 83 mg/mL, respectively) [45,46]. The maximum levels reached after 40 h were 70 and 60% of the dose for verapamil and for rifampicin, respectively. The drugs profiles reflected their relative solubility in water as well as their degree of encapsulation. The mass balance with the active ingredients remaining in the donor and on the membrane accounted for 100% of the added drugs.

### 3.9. In Vitro Antimicrobial Activity against M. tuberculosis

As a further step, we evaluated whether the powder formulations could afford a synergistic MICs restricting the *M. tuberculosis* growth in vitro. The test was performed in order to analyze what was the killing activity of the formulation on different Mtb strains in presence and absence of verapamil (Table 4). Both powder A (PA) and B (PB) showed antituberculosis activity with a combined MIC_99_ of 0.25 µg/mL for the susceptible strain H37Rv and a combined MIC of 32 µg/mL for the two drug resistant strains. As expected, hyaluronic acid showed no inhibitory effect on *M. tuberculosis* growth at the concentrations used in the powder formulations. No difference was observed in vitro between the formulation containing verapamil (PA) and the formulation with the antibiotics alone (PB) for all strains. We hypothesized that the system developed could be used at non-cytotoxic levels and simultaneously be highly active against *M. tuberculosis* inside the macrophage, due to the concurrently presence of a polymer able to increase AMs uptake and an efflux pump inhibitor. Thus, the cytotoxicity against macrophages was first of all evaluated.

### 3.10. Cytotoxicity Evaluation

Cytotoxicity assays showed that the solutions obtained from SD powder A (Powder A concentration 89.5 µg/mL, with verapamil-RIF 31.3 µg/mL; INH 16.1 µg/mL; VER 10.7 µg/mL as baseline; HA 31.4 µg/mL) and SD powder B (PB concentration 82.5 µg/mL, without verapamil-RIF 31.4 µg/mL as baseline; INH 19.8 µg/mL; HA 31.4 µg/mL) were non-toxic to human macrophages, maintaining 100% macrophage viability at the concentrations used in the infection assays after 3 days of exposure (Figure 8). At day 5, macrophage viability remained at 100% in the presence of PA and PB, starting to decrease at day 7 reaching 12% macrophage viability in the presence of PA and 64% of viability in the presence of PB. At day 9, viability is maintained at 12% in the presence of PA, whereas for PB viability decreased to 34%. These results confirmed that powder formulations containing verapamil at concentrations equal to 10 µg/mL do not compromise macrophage viability after 5 days of exposure. The solution of Powder A and Powder B, had concentrations that are above the average serum concentration of these drugs used in clinical practice (0.15–0.4 µg/mL, VER; 10 µg/mL RIF, 5–10 µg/mL INH [47,48]. 

The testing of high concentrations of anti-tuberculosis drugs intended to assess if a higher concentration of both isoniazid and rifampicin could be reached inside the macrophage with minimal toxicity effects due to the nanoparticle encapsulation and simultaneously have an antimicrobial effect on the multi/extensively drug resistant strains. Therefore, we tested the platform and the biological system at its upper limit to assess the spectrum efficacy, from fully susceptible to highly resistant *M. tuberculosis* strains (Table 4) and simultaneously evaluated the associated toxicity (Figure 8). 

### 3.11. Ex Vivo Assay

To confirm the hypothesis that the SD powder containing the three drugs combination could restrict the *M. tuberculosis* growth inside human-monocyte derived macrophages, the intracellular activity against *M. tuberculosis* of the powder formulations was tested using as control a simple solution of the drugs (non-encapsulated). As shown in Figure 9, both formulations were able to kill intracellular *M. tuberculosis* promoting more than 80% reduction in bacterial viability irrespective of the drug resistance profile at day three post-infection. No significant difference was observed between the formulation containing verapamil compared to that not containing verapamil, most probably because we tested very high doses of the drugs. A dose optimization was not performed.

## 4. Discussion

A significant part of the current research on new formulation strategies for the treatment of lung infections including MDR-TB is oriented toward the elimination of the use of excipients, with the aim to maximize the amount of drug delivered and the drug concentrations at the site of action [49,50,51].

In this work a different *modus operandi* is proposed, since a relatively large amount of excipient (HA) (35% *w*/*w*) was used to build a respirable powder. The rationale was to put together two different strategies to increase the drug concentration within the infected macrophages, evaluating this strategy through indirect observations, i.e., drugs release profiles and ex vivo activity in infected AMs. The first strategy aimed at increasing the antibiotic uptake by AMs. In fact, the use of nanoparticles with suitable size (around 400 nm), as well as the composition, i.e., low molecular weight HA, which should improve the AMs recognition through CD44 membrane receptor binding inducing the particles internalization. The second strategy applied was to increase of the persistence of the antibiotics within the macrophages as well as into the mycobacteria, through the addition of verapamil as efflux inhibitor. As already stated, verapamil has been shown to accelerate bacterial killing in mice infected with normal and drug-resistant Mtb strains [52]. In the present work, the administration of the efflux inhibitor directly to the lung along with an antibiotic drugs combination is proposed to significantly reduce the risk of systemic cardiovascular side effects [53], which are usually reported for oral and systemic delivery of verapamil [54]. Furthermore, early studies evidenced its safety by inhalation in a relatively large dose range, in particular in non-asthmatic subjects [53,55,56]. 

Highly inhalable aggregates of HA nanoparticles encapsulating the two antibiotics and verapamil were produced by spray drying. Despite the irregular shape, the microparticles produced presented very small dimension (*D*_v,50_ 0.94 μm) particularly affecting the high respirability. The produced SD powder, did not presented the peaks in the PXRD patterns relevant to a crystal structure while it showed the typical halo of an amorphous material even if all the three raw drugs were crystalline in nature.

Despite the relatively high residual water content, aerodynamic characteristics of the powder were not influenced by storage at room temperature for one month. Additionally, verapamil proved to be stable in all conditions tested. The isoniazid content reduction detected at high temperature can be explained considering a potential degradation ascribable also to the above-mentioned residual water in the powder [57].

The relatively low encapsulation rate of isoniazid was attributed to its hydrophilic nature (solubility in water 140 mg/mL) [58] and the production method of the nanosuspension. Indeed, isoniazid was dissolved in water together with the polymer; so, during the formation/precipitation of the polymeric nanoparticles, the drug tended to remain into the aqueous phase, resulting in a relatively low entrapment. Dissolution of the isoniazid in the alcoholic fraction may be an approach to improve its encapsulation efficiency in the polymer nanoparticles. In general, encapsulation efficacy values were slightly lower than those reported in other formulations using hyaluronic acid [59,60], but equal, and in some cases even higher, to the data published for nanoparticles prepared using the same drugs separately and a different type of polymers, such as chitosan [61,62,63,64]. 

The better aerodynamic performance, obtained with Turbospin^®^ device, can be explained taking into consideration the relatively high stickiness of the powder stemming from the high residual water content. With such a powder, the emission mechanism of this device, which implied a reduced contact between the powder and the plastic parts of the device resulted in a higher emitted dose [65].

The high residual water content represents a weak point of the formulation that may be addressed by storing the powder in a dry environment for a suitable time after the production.

The rationale beyond the production of respirable powder starting from a nanosuspension implies the reconstitution of the parent nanoparticles once the microparticles get in contact with the biological fluids. Because of their size and composition, the nanoparticles would then be more prone to capture and internalization by AMs. The de-aggregation kinetics data, obtained by employing magnetic stirring/ultrasound application, suggest that, once inhaled, these microparticles are be able to de-agglomerate slowly in about one or two hours, giving rise progressively to particles with a diameter suitable for AMs phagocytosis [25,27]. 

The amount of antibiotics taken by the infected AMs is influenced by the size and composition of particles produced. In this respect, the release rate of the drugs from the particles has to be taken into consideration as a key feature for achieving an optimal efficacy since the drugs should not be release significantly before the particle uptake from the AMs. The time for complete AMs uptake has been reported to occur within 1 to 6 h [66]. 

For verapamil and rifampicin the release kinetics was more linear compared to isoniazid, which showed the typical release profile of a water soluble molecule from an hydrophilic matrix [67]. It is worth underlying that the relative release rate of the three drugs mirrors the relevant solubility in water as well as their degree of encapsulation and that the mass balance with the active ingredients remaining in the donor and on the membrane accounted for 100% of the added drugs.

Although the dissolution test was performed in sink condition for all drugs (as also demonstrated by the liner profile of the drugs with lower solubility), the release rate for each drug was significantly lower compared to the data reported in the literature. For isoniazid the value reported in other studies was 60% of the dose released in 20 min [37]; in another case a nearly complete dissolution (90%) was reported in 8 h [68]. For the less hydrophilic drugs, rifampicin and verapamil, the time reported to reach the maximum level of dissolution was within 3–4 h (70–90%) [37,69,70]. 

We explained these discrepancies by considering that these data were collected using different apparatuses, dissolution media and drug doses. Here, a Franz type cell with limited liquid volume in the donor compartment was adopted as dissolution method instead of the compendial USP Apparatus I, II or IV in order to keep the in vitro dissolution conditions closer to those present in the lung. In fact, the method adopted here, is one of the few capable to create an air-liquid interface which somehow resemble to what happens on the lung inner surfaces [71]. 

Therefore, the obtained de-aggregation and dissolution data strongly support the hypothesis that once in the lung the spray dried particles will release nanoparticles lasting for a time sufficient for AM uptake. It is also worth underscoring that the slow, but progressive, release of the antibiotics will be crucial to maintain high drug levels inside AMs [72].

Antimicrobial activity results showed that the powder formulation is safe for macrophages and is effective against drug susceptible and drug resistant *M. tuberculosis* in vitro. Furthermore, an enhancement of the antimicrobial activity of the two standard antituberculosis drugs—isoniazid and rifampicin—was evidenced particularly for the efficacy against the M/XDR strains. Noteworthy, both encapsulated powder formulations promoted significant reductions in the number of intracellular drug resistant strains, allowing the delivery of high doses of antibiotics with minimal toxic effects even in a prolonged exposure (more than three days), confirming the usefulness of this approach against all forms of Mtb for prolonged therapeutic regimens. Regarding the non-encapsulated formulations, no difference was observed with both drug formulations for the susceptible H37Rv strain, as expected, due to its reduced efflux activity [17] combined with the high concentrations of isoniazid and rifampicin used (see Table 4 for MICs). For the resistant strains, it was observed a modest increase in the number of viable bacteria in the absence of verapamil due to their well described increased efflux activity [42]. These data indicate that the synergistic effect of the efflux inhibitor verapamil is dependent on the efflux activity of the strains. 

Additional work should focus on concentration(s) optimization of each antibiotic in the powder and the combination of the antibiotics and verapamil, in order to further reduce the resistance of the MDR and XDR strains with the aid of an efflux inhibitor, maintaining all compounds at clinically achievable concentrations with minimal toxicity associated.

## 5. Conclusions

In the present work we have developed and tested a new platform for lung delivery of antibiotics aiming at efficiently treating mycobacterial infections and tackling the multidrug resistance phenomenon.

In the case of TB, the efficient administration of antibiotics to the lung implies the capability to reach the alveolar region with particles of suitable aerodynamic diameter as well as to deliver the drugs into the AMs where the Mtb resides. We addressed these issues by exploiting the peculiar characteristic of HA sodium salt in aqueous solution to produce, upon treatment with an antisolvent such as ethanol, to nanoparticles with size suitable for AM uptake. An HA low molecular weight to further improve the AMs recognition and engulfment was selected. The spray drying of the nanosuspension affords a powder constituted of microparticles with a mean volume diameter of 1 μm, ideal for a deep lung deposition. These microparticles are constituted of agglomerates of nanoparticles, that restore their original nanosize upon contact with water and release their drug content in a sustained manner.

To further improve the drug efficacy, the particles contain an efflux inhibitor to maintain the suitable dose of the drug into both the macrophages and Mtb. 

The formulations were effective in vitro and ex vivo against both susceptible and resistant Mtb strains and demonstrated to maintain the same efficacy and low toxicity after three days post- infection. The main advantage of the encapsulated formulations is that they can reach, deposit and concentrate in the lung macrophages high doses of antibiotics formulations without the associated cytotoxicity and adverse effects, especially for long-time therapeutics and when used in aerosolized formulations. The presence of an efflux inhibitor like verapamil enhances the killing activity and protect antibiotics that are prone to be effluxed. Future work will involve drug concentration optimization for maximum efficacy with the lowest toxicity possible, to be further validated ex vivo and confirmed in vivo (animal model).

Obviously, we do not present here data that have to be used in a dossier for the regulatory approval of a new medicinal product, however the data collected, do represent a first, yet strong, scientific evidence of the suitability of the adopted approach to enhance the macrophage elimination of Mtb, irrespective of its drug-sensitiveness profile, a strategy to be further explored against respiratory M/XDR *M. tuberculosis* and NTM infections. Being AMs the primary target of the formulation, the level of drug resistance becomes less relevant and the powder formulation appropriate to treat any mycobacterial infection.

## Figures and Tables

**Figure 1 pharmaceutics-11-00203-f001:**
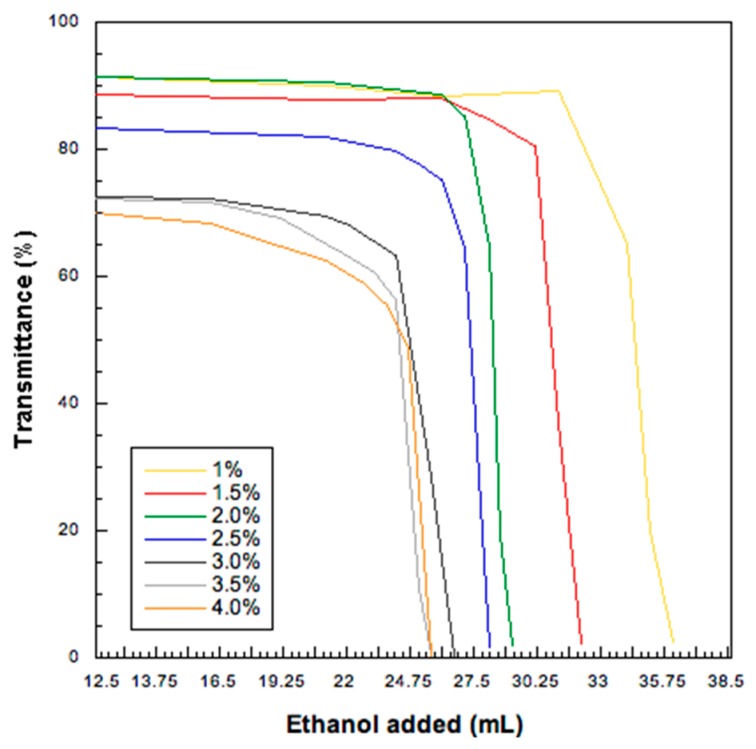
Transmittance values (%) as a function of ethanol volume added for six sodium hyaluronate aqueous solutions at different concentration concentrations (starting HA concentration from 1.0 to 4.0% *w*/*v*.

**Figure 2 pharmaceutics-11-00203-f002:**
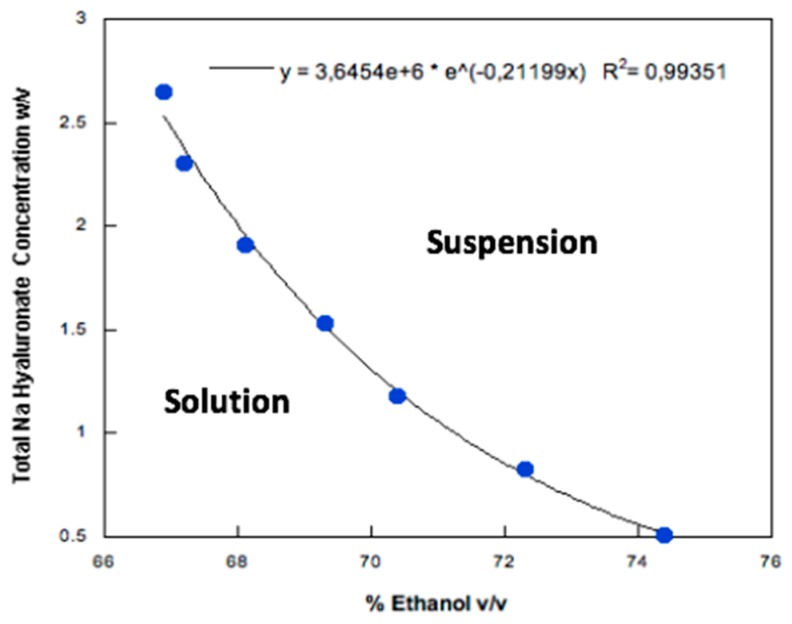
Sodium hyaluronate concentration in a water/ethanol mixture as a function of the ethanol volume necessary to obtain the inflection point in turbidimetric analysis.

**Figure 3 pharmaceutics-11-00203-f003:**
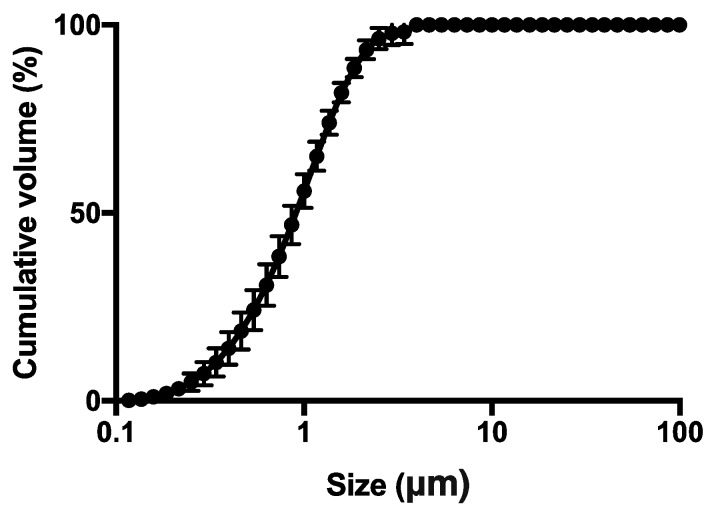
Particle size cumulative undersize distribution evaluated by laser light diffraction of the spray dried powder (square). The bars represent the standard deviation (*n* = 3).

**Figure 4 pharmaceutics-11-00203-f004:**
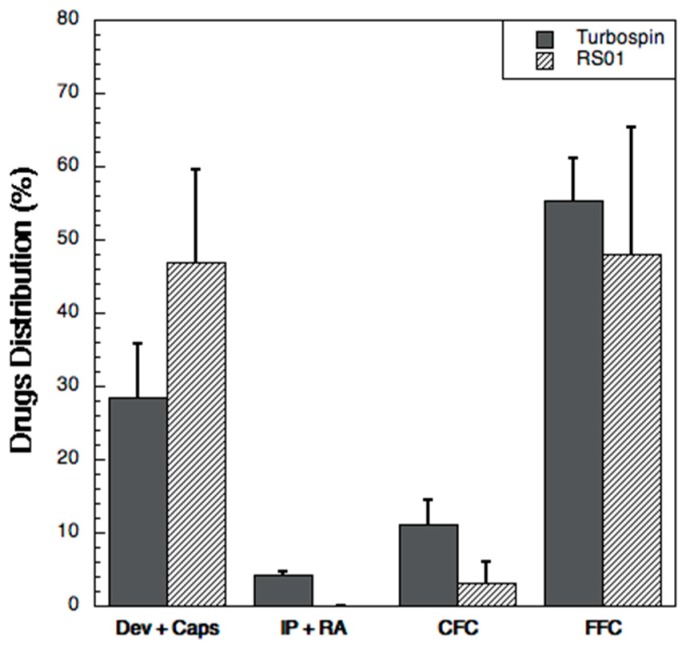
Drug distribution (% of the loaded amount) of the three drugs (sum of them) upon aerosolization with RS01^®^ and Turbospin^®^ in the Fast screening impactor: device and capsule (Dev + Caps), induction port and rubber adaptor (IP + RA), coarse fraction collector (CFC) and fine fraction collector (FFC). Data are expressed as mean of the values obtained for each drug in each analysis; the bars represent the standard deviation (*n* = 3).

**Figure 5 pharmaceutics-11-00203-f005:**
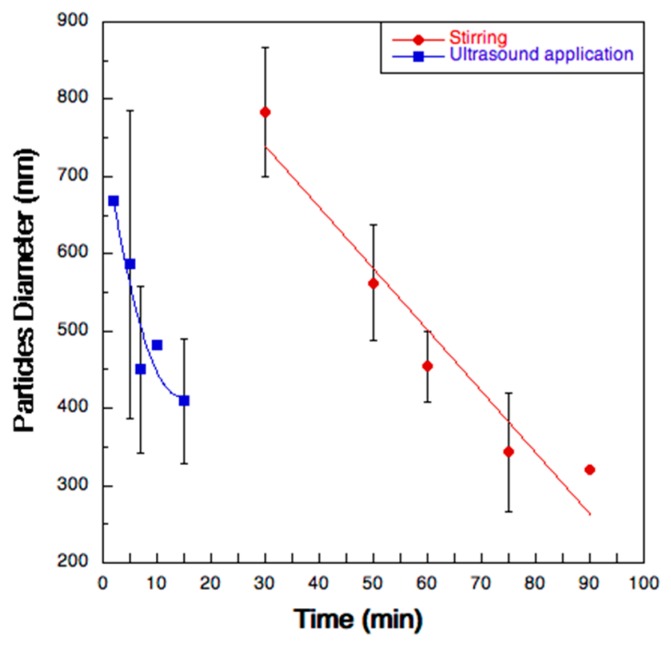
Diameter variation as a function of time of the microparticles suspended in two different test conditions: magnetic stirring (red circles) and ultrasound application (blue squares).

**Figure 6 pharmaceutics-11-00203-f006:**
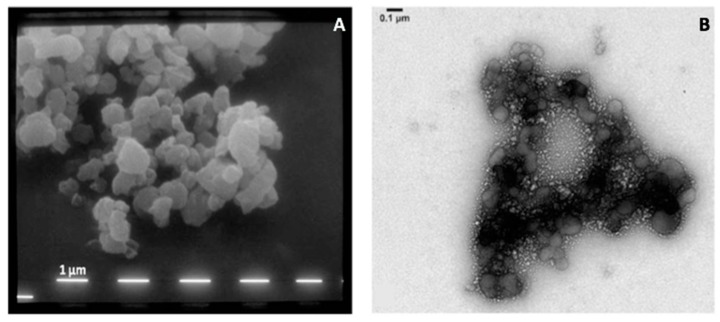
SEM (**A**) of the microparticles constituting the powder and TEM (**B**) pictures of the nanoparticles agglomerated to form the microparticles.

**Figure 7 pharmaceutics-11-00203-f007:**
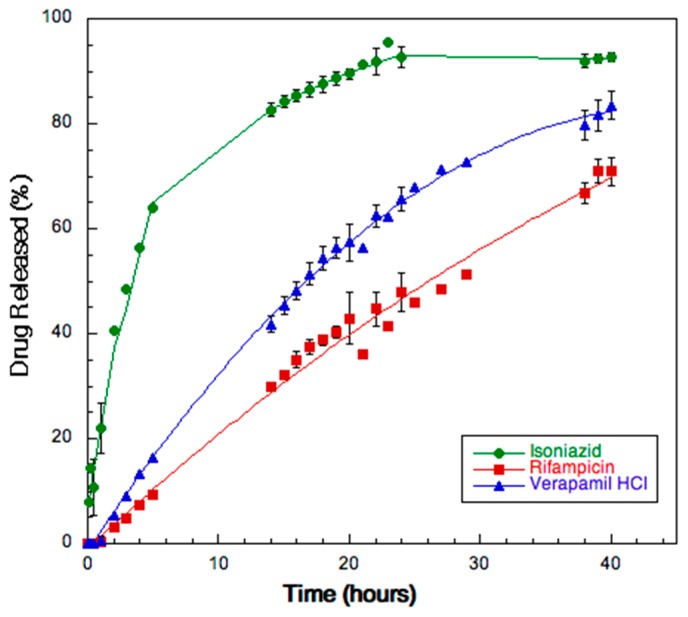
Release profiles of the three drugs from SD powder A.

**Figure 8 pharmaceutics-11-00203-f008:**
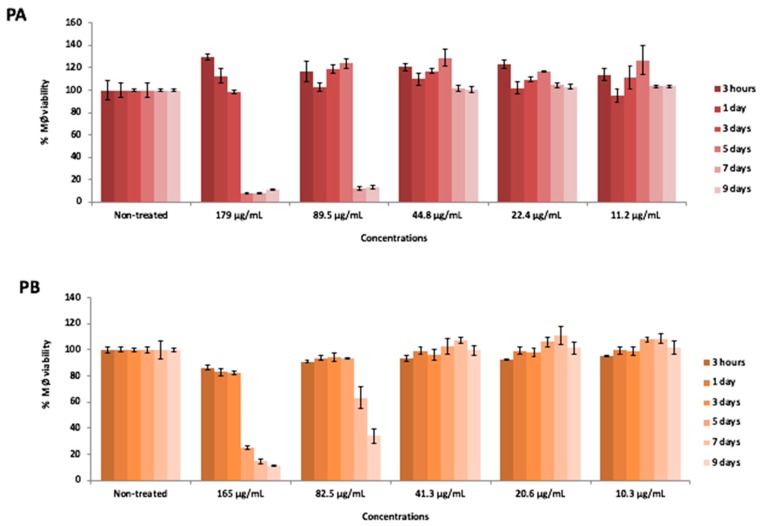
Macrophages viability, expressed as percentage respect to the control, after different times of incubation (3 h or 1, 3, 5, 7 and 9 days) with five distinct solutions prepared from SD Powder A (PA) and SD Powder B (PB) at different concentrations.

**Figure 9 pharmaceutics-11-00203-f009:**
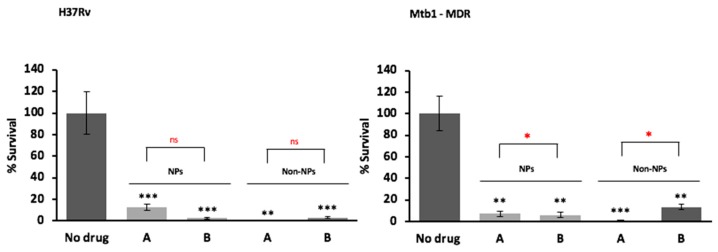
In vitro antimicrobial activity of solutions obtained from dried powder (A) and an analogous dried powder without verapamil (B) against drug susceptible (H37Rv) and drug resistant Mtb strains (M/XDR), as nanoparticles formulation (NPs) or as free solution of the drugs without HA (Non-NPs) at day 3 post-infection. Values are expressed as mean of triplicate and bars represent standard deviation (ns = not statistically significant; * significant *p* < 0.05; ** highly significant *p* < 0.01). Powder A (NPs)—rifampicin 31.3 µg/mL; isoniazid 16.1 µg/mL; verapamil 10.7 µg/mL (as baseline); low molecular weight HA 31.4 µg/mL; powder B (NPs)—rifampicin 31.4 µg/mL (as baseline); isoniazid 19.8 µg/mL; low molecular weight HA 31.4 µg/mL. Solution A (Non-NPs)—rifampicin, 30 µg/mL; isoniazid, 15 µg/mL; verapamil, 10 µg/mL (as baseline); solution B (Non-NPs)—rifampicin, 30 µg/mL (as baseline); isoniazid 17.6 µg/mL. The levels of significance were calculated comparing each of the mixtures with the control with no drug denoted in the graph by black * and by comparing each mixture encapsulated with the non-encapsulated formulation symbolized in the graph by the red *.

**Table 1 pharmaceutics-11-00203-t001:** Drugs content of the SD powder (mean value and standard deviation) at time zero and after storage for 1 month at 25 °C and for 24 h under stressed condition (50 °C).

Drug	Nominal (% *w*/*w*)	Assay (% of the Nominal Value)		
		Time zero	Month 1/25 °C	24 h/50 °C
Isoniazid	18	98.4 ± 0.03	90.4 ± 0.02	81.3 ± 0.02
Rifampicin	35	100 ± 0.03	100 ± 0.01	96.9 ± 0.03
Verapamil	12	100 ± 0.02	100 ± 0.01	99.2 ± 0.04

**Table 2 pharmaceutics-11-00203-t002:** Emitted Dose (ED), Emitted Fraction (EF), Fine Particle Dose (FPD) and Fine Particle Fraction (FPF) obtained upon aerosolization with the two devices, referred to the single drugs (*n* = 3 ± standard deviation).

Drug	RS01^®^				Turbospin^®^			
	ED (mg)	EF (%)	FPD (mg)	FPF (%)	ED (mg)	EF (%)	FPD (mg)	FPF (%)
Isoniazid	0.44 ± 0.14	49.35 ± 15.32	0.40 ± 0.16	90.62 ± 8.15	0.63 ± 0.17	88.72 ± 3.35	0.47 ± 0.10	76.88 ± 7.48
Rifampicin	0.87 ± 0.26	47.86 ± 14.19	0.83 ± 0.29	93.55 ± 5.69	1.23 ± 0.20	74.75 ± 6.23	0.96 ± 0.15	77.68 ± 4.87
Verapamil	0.31 ± 0.10	50.89 ± 16.16	0.30 ± 0.11	93.89 ± 5.72	0.37 ± 0.04	68.43 ± 3.14	0.30 ± 0.05	80.54 ± 5.01

**Table 3 pharmaceutics-11-00203-t003:** Emitted Dose (ED) and Fine Particle Dose (FPD) of powder aerosolized by RS01^®^ during stability study, after 1 month at 25 °C and after 24 h under thermal stress (50 °C) (*n* = 3 ± standard deviation).

Drug	RS01^®^							
	1 Month/25 °C				24 h/50 °C			
	ED (mg)	EF (%)	FPD (mg)	FPF (%)	ED (mg)	EF (%)	FPD (mg)	FPF (%)
Isoniazid	0.50 ± 0.05	61.99 ± 5.92	0.46 ± 0.04	92.34 ± 1.28	0.42 ± 0.01	58.75 ± 1.96	0.37 ± 0.01	87.95 ± 0.57
Rifampicin	1.03 ± 0.11	58.56 ± 6.07	0.93 ± 0.08	90.81 ± 1.21	0.95 ± 0.06	55.81 ± 3.11	0.80 ± 0.05	83.62 ± 0.83
Verapamil	0.37 ± 0.03	62.83 ± 5.35	0.34 ± 0.03	92.26 ± 0.85	0.35 ± 0.02	59.54 ± 4.24	0.31 ± 0.03	88.81 ± 1.33

**Table 4 pharmaceutics-11-00203-t004:** Minimum inhibitory concentrations (MIC) of the dried powder (A), a spray dried powder without verapamil (B) and the components of the formulation taken individually (rifampicin, RIF; isoniazid, INH; verapamil, VER; low molecular weight sodium hyaluronate, HA). The assays were performed at least in triplicate (biological replicates).

Specimen	MIC (µg/mL)		
	H37Rv INH^S^/RIF^S^	Mtb1-MDR	Mtb2-XDR
SD Powder A *	0.25	32	32
SD Powder B **	0.25	32	32
RIF	0.5	512	512
INH	0.1	12.8	12.8
VER	512	512	512
HA	>256	>256	>256

* SD Powder A weight %: RIF 35; INH 18; VER 12; HA 35. ** SD Powder B weight %: RIF 38; INH 24; HA 38.

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
