# Peer review of "Sodium Hyaluronate Nanocomposite Respirable Microparticles to Tackle Antibiotic Resistance with Potential Application in Treatment of Mycobacterial Pulmonary Infections"

_pharmaceutics, 2019, doi:10.3390/pharmaceutics11050203_

Reviewer 1 Report

This manuscript describes the formulation, characterization and in vitro/ex vivo testing of anti-TB drugs along with an efflux pump inhibitor (EPIs), verapamil. The studies were evaluated using 3 Mtb strains using human monocytes, which makes this study important. This is a novel dry powder inhaled approach to treat tuberculosis. Based on the threat of drug resistance against most (if not all) of the first and second line TB drugs, such an approach using EPIs is urgently required. The authors should consider the below suggestions.

Author Response

General comment

The authors discuss the role inhaled EPIs, along with anti-TB drugs, that will play a role against NTM diseases. However, their studies only evaluate Mycobacterium tuberculosis (drug susceptible and resistant strains), and not NTM. The authors might want to focus on MTB in the introduction.

We have better focused on Mtb in the introduction.

The authors mention that the spray drying conditions were used from another published study (ref 38). However, was the spray drying parameters optimized for this study (HA, verapamil, Rif and INH)?

The spray drying process was not optimized for this study, the only process optimized was   the  nanoparticle formation and components content (see Supplementary material, Figure A).

The authors showed that high moisture content in the spray dried powders had adversely impacted powder characterization. Would spray drying at higher inlet (and subsequently higher outlet) temperature allowed for lower moisture in the dry powders? Please discuss.

- The employment of the spray drying at higher inlet temperature would be definitely be a good procedure to reduce the residual moisture content in the product, and it is commonly adopted when possible.

However, in the specific case it was not possible to increase the inlet temperature because the suspending liquid of the suspension we died contained 70% v/v of ethanol. The drying at a higher temperature would only be possible employing a specific apparatus (Inert Loop). Hence for safety reason different approaches were studied in order to reduce the moisture content after the process (not reported in the paper), namely the dried powder was put in a vacuum desiccator for 5 days; thereafter, the residual moisture content obtained was half of the one at time zero (about 3% w/w).

Although verapamil (EPI) concentration used in the studies were not toxic to AMs after 5 days exposure, is this concentration sufficient to act as an EPI? If a dose optimization was not performed, please mention this as a limitation to the study.

A dose optimization was not performed. This was added in the text as a new sentence. However, as reported in literature, a verapamil concentration of few mg/mL is enough to exert efflux pump inhibition.

Specific comments

Abstract

Line 22- “other mycobacterial species are..”

Done

Line 28- “these particles were able to regenerate their original size upon contact with aqueous environment”. However, the original size was achieved using magnetic stirring/sonication. If this is true, please mention this here.

Done

Introduction

Line 68- “activity of the macrophage….”

Done

Line 69- Does verapamil impact the efflux pumps in AMs (mammalian cells) also, in addition to mycobacterial species?

This point is addressed in the cited reference 13

Line 110- “to work regardless of the profile…”

Done

Line 110- Not sure if the last statement is true? EPIs can only work against drug-tolerant Mtb? Once the bacteria become resistant (irreversible gene mutations), EPIs usually do not work. Please clarify.

It is correct, EPIs can only work against sensitive species. We have clarified in the text.

Materials and methods

Line 159- Did the authors look at the remaining drug in the nanoparticles, beyond looking in the filtrate?

A mass balance was performed.

Line 275-76- The authors mention that sampling was done every hour for 40 hours? However, Figure 7 does not reflect time points for each hour.

Some hours were missing due to the extension of the experiment over-night. However, we tried to cover as much hours as possible in the replicates of the experiment. We modified the text to account for the Reviewer observation

Line 316- “5X 10^4 cells were…” Done

Line 317- Did the powder formulations remain in the cell culture for 1, 3, 5 and 7 days? Or were the formulations removed after a few hour exposure? If the particles were not washed, it could impact the fluorescence measurement readings. Please clarify.

We added a sentence to clarify the point.

For sections on cytotoxicity analysis and ex vivo assays, how were the microparticles exposed to the cell culture? Please mention.

We specified in the text that how the particle were suspended in the cell medium.

Line 334, 337, 340, 341- what does baseline mean?

For  powder A (nanoparticles formulation, NPs) – rifampicin, 30 µg/ml; isoniazid, 15 µg/ml; verapamil, 10 µg/ml (as baseline); low molecular weight HA, 30 µg/ml;

Verapamil has to be used at concentrations that are nontoxic for primary macrophages, 10 μg/ml (Machado et al., 2016, PloS One). Therefore, the nanoparticles formulation corresponding to power A was prepared to contain 10 μg/ml of VP. Since each formulation has a defined amount of each component, the concentrations of rifampicin, isoniazid and HA present in this mixture were determined based on the amount of powder used to prepared the solution containing verapamil as baseline. Taking in consideration a solution prepared to contain verapamil at 10 µg/ml (as baseline), the concentration of rifampicin, isoniazid and low molecular weight HA were determined as 30 µg/ml, 15 µg/ml and 30 µg/ml, respectively.

For powder B (nanoparticles formulation NPs) – rifampicin, 30 µg/ml (as baseline); isoniazid 17.6 µg/ml; low molecular weight HA 25.6 µg/ml);

This suspension (without verapamil) was prepared using the same rational described above, however using rifampicin as baseline, prepared to contain the same concentration tested in the formulation with verapamil.

For  solution A (free solution mixtures, Non-NPs) – rifampicin, 30 µg/ml; isoniazid, 15 µg/ml; verapamil, 10 µg/ml (as baseline);

This solution was prepared to contain exactly the same final concentrations used in powder A, but not encapsulated)

For  solution B (free solution mixtures, Non-NPs) - rifampicin, 30 µg/ml (as baseline); isoniazid 17.6 µg/ml.

This solution was prepared to contain exactly the same final concentrations used in powder B, but not encapsulated).

Results

Line 410- Did the low encapsulation efficiency for INH (53.93%) not impact the actual drug loading? Table 1 shows the same actual loading to that of theoretical loading.

Table 1 reports the actual content of drugs in the spray dried powder regardless the fact that is is encapsulated or not. The non-encapsulated INH is anyhow present in the powder in form of free solid microparticles.

Figure 4- Is this data shown after adding all the 3 drug values?

Yes, it is, as reported in figure caption “Data are expressed as mean of the values obtained for each drug in each analysis”.

Table 2- Why is higher variation (between 88.72- 68.43%) observed with Turbospin EF (column 6) if the drugs in the NPs were at the same ratio? Powder aerosolization using the inhalers into the FSI should keep the drug concentration ratio same.

Data were calculated based on the actual amount of the drugs present in the powder loaded in the capsule (5 mg). The non-completely uniform distribution of the different components (encapsulated or not) likely affected the aerosolization behavior.

Line 514- “this was not surprising…” This statement by the authors is not clear. The authors wanted to see a difference in antimicrobial activity based on the first sentence of the para. Please reframe this sentence.

Done.

Line 516- Why do the authors feel that the combination therapy will only be active against Mtb inside the macrophage? Please clarify.

Done.

Table 4- Does this show that INH alone had better efficacy against all 3 Mtb strains? If so, please discuss why.

Not completely. Both clinical strains presented high-level resistance to rifampicin (512 μg/ml) and isoniazid (12.8 μg/ml) and both have mutations conferring resistance to both antibiotics. What we are seen it is the synergy between each drug in the formulation. We assume that in the mixtures the MIC could not be lowered below 32 μg/ml due to high MIC of rifampicin that however could be lowered 16x when combined with isoniazid. The effect of the combination of the MIC of isoniazid is not observed since its initial MIC is lower than 32 μg/ml, but it could be lower also. Nevertheless, this effect could only be tested using a strain resistant to isoniazid but susceptible to rifampicin.         

Figure 9- Is the statistics (level of significance) compared with ‘no drug’ group?

We added a sentence in the figure legend to clarify this point

Discussion

Line 591-596- The authors mention two strategies- first, the increased uptake of antibiotic by AMs using LMW HA. However, they did not study this claim that low molecular wt. HA allowed higher AM uptake via CD44 receptors. Even the second strategy “persistence of antibiotics within the macrophages as well as mycobacteria, through the addition of verapamil…” was not studied. Please modify these statements to reflect indirect evaluation of the two rationale. Done.

Line 628-629- The authors state the MPs will de-aggregate in one or two hours after deposition in the lung. However, the authors achieved de-aggregation after magnetic stirring/ultrasound. Please clarify.

Done.

Line 639-640- Please mention the mass balance data in the results section also.

Done.

Reviewer 2 Report

In the present work, authors proposes a formulation approach for the treatment of mycobacterial infections potentially suitable to overcome antibiotic resistance. They developed a respirable particles (micron-sized aggregates loaded with two different antibiotics, rifampicin and isoniazid, and one efflux pump inhibitor, verapamil) for lung delivery of antibiotics aiming at efficiently treating mycobacterial infections and tackling the multidrug resistance.

Overall the manuscript looks good. However, minor revision is required before considering for publication.

·         The introduction and the conclusion part need to be revised.

·         The Figure, figure legends and tables are not prepared well and need to be improved from its original form.

·         Materials and methods section require extensive revision.

·         This manuscript required typographical and grammatical correction.

Author Response

We would like to thank the Reviewer for the positive comments.

·         The introduction and the conclusion part need to be revised.

We made a significant revision of the introduction and conclusion

·         The Figure, figure legends and tables are not prepared well and need to be improved from its original form.

We modified the Figure legends

·         Materials and methods section require extensive revision.

We extensively revised the material and methods part by adding more detailed description of the method adopted.

·         This manuscript required typographical and grammatical correction.

We fixed the topographical and grammatical errors

Reviewer 3 Report

This study reports the formulation of a respirable hyaluronic acid microparticles for treatment of intracellular M. tuberculosis infection and to possibly overcome the antibiotic resistance. The major goal was to design and develop highly respirable particles with enhanced alveolar macrophages (AMs) uptake that could provide increased intracellular anti-tuberculosis drug concentrations. They achieved this by designing sub-micron particles (300-500 nm) of two antituberculosis drugs, isoniazid and rifampicin and verapamil which is an efflux pump inhibitor which could impair drug efflux. They combined this with sodium hyaluronate salt (HA) that increases capture of the particles by AMs. The authors completed physicochemical characterization, antimicrobial activity testing, in vitro and ex-vivo studies of the microparticles.

Since one of the targets of the paper is to address drug resistance to anti-tuberculosis drugs, the authors should include, in the introduction, a brief description of how INH and RIF resistance arises. This would assist the reader to make a judgement on whether the proposed approach (by the authors) is likely to overcome this resistance.

In all centrifugation processes, there is no mention of the temperature conditions stipulated. Please include this detail.

The size and shape of particles is important for pulmonary delivery. The authors should comment on how the shape of their particles could affect the deposition of the particles into the alveoli or emission from the breathing devices?

The authors should describe any possible approaches to improve the entrapment efficiency of isoniazid.

There are some grammatical errors that need to be corrected. Some sentences are not expressed in same tense as some words. This changes the meaning.

Author Response

Since one of the targets of the paper is to address drug resistance to anti-tuberculosis drugs, the authors should include, in the introduction, a brief description of how INH and RIF resistance arises. This would assist the reader to make a judgement on whether the proposed approach (by the authors) is likely to overcome this resistance.

 We added a sentence in the introduction to clarify this point.

In all centrifugation processes, there is no mention of the temperature conditions stipulated. Please include this detail.

Done.

The size and shape of particles is important for pulmonary delivery. The authors should comment on how the shape of their particles could affect the deposition of the particles into the alveoli or emission from the breathing devices?

 We added a sentence in the discussion on this point

The authors should describe any possible approaches to improve the entrapment efficiency of isoniazid.

  We added a sentence in the discussion on this point

There are some grammatical errors that need to be corrected. Some sentences are not expressed in same tense as some words. This changes the meaning.

We corrected the grammatical errors in the text.